# Reproducibility report of Learn to Resolve Conversational Dependency: A Consistency Training Framework for Conversational Question Answering paper

## Reproducibility Summary

**Scope of Reproducibility**

In this process, in order to evaluate the accuracy of the claims mentioned in the paper and also the reusability of the paper codes, the codes introduced in GitHub were used. For this purpose, it was carried out in accordance with the instructions mentioned in it. Due to severe hardware limitations, it was not possible to learn the model and re-implement the code using Google Colab.

**Methodology**

Contrary to what was mentioned in the article about executable hardware, Google Colab was used with the following specifications: GPU 13GB RAM and 80GB Disk were used.

The duration of the model evaluation process on the Google Colab with the mentioned features is approximately 18 minutes and 30 seconds, and the disk and GPU consumed in this process are 49GB and 4GB, respectively.

**Results**

In the evaluation performed using the proposed RoBERTa model of the paper, the criterion F1 67.73891723166825 was obtained, which is quite similar to the accuracy reported by the paper itself.

**What was easy**

The hardware requirements and the initial setup of the experiment were fully described in the paper in Section B (Hyperparameters), which was very helpful in re-executing the code. A description of all usable datasets was also provided in Section 4.1 (Datasets). The documentation published at the Git by the authors was almost comprehensive and practical, include installation requirements, hardware, data sets, how the model is taught and how the model is evaluated.

**What was difficult**

The authors used a 24GB GPU (RTX TITAN). Execution with such conditions is not possible due to the free features provided by Google Colab. Due to the mentioned limitation, we tried to change the batch size, which was set to 12 by default in the article, to 2; But we still had a lack of RAM from Colab. It should be noted that by reducing the batch value, we also changed the number of epochs, but there were still problems.

**Communication with original authors**

Due to the comprehensive documentation provided in the gate as well as in the text of the article, there was no need to interact with the authors. Of course, the gateway account and the authors' research gate account were available in ways to communicate with the authors, including email.

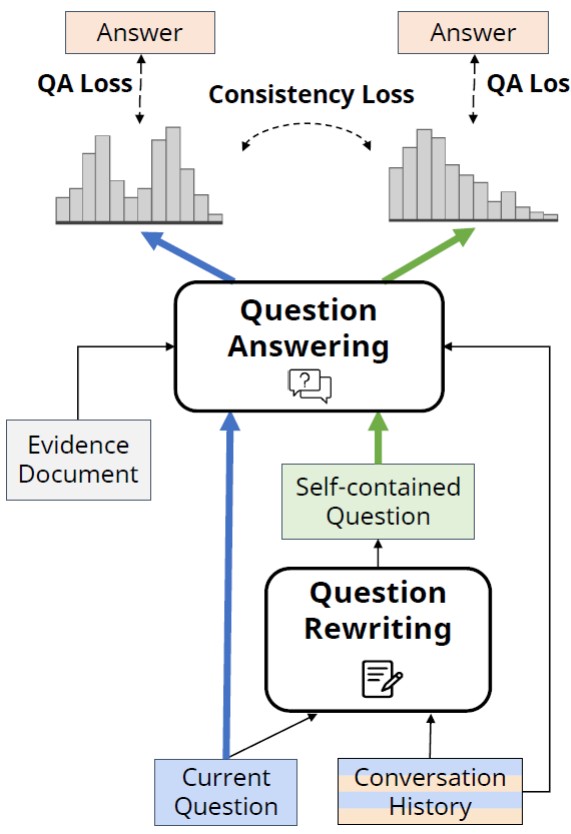

Figure 1: Proposed framework

## 1  Introduction

A Conversational Question Answering (CQA) is the design of an intelligent conversation system that can not only engage in an interactive conversation at the human level, but also go beyond it and answer questions on a variety of topics, one of which is today Prominent goals in the field of artificial intelligence. Conversational AI is an integral part of natural user interfaces. The main idea of the CQA is to ask the machine to answer a question based on the part provided. The approach to solving these problems is proposed: the end-to-end approach and the pipe line approach. This article uses the pipeline approach.

In this paper, using a question and pipeline method, we have tried to teach a question and answer model that is able to answer conversational questions. RoBERTa was trained using the ExCorD framework. This model has been taught once and, according to the authors, can be used to evaluate common conversational Q&A models. In order to assess the feasibility and potential of code re-implementation and to discuss the method presented in the article, the maximum free hardware features of Google Colab were used.

## 2  Scope of reproducibility

In this section, the claim made by the authors of the article is examined. In the pipe line method, to generate ambiguity questions from the main questions, if the T5 model is used to prepare a set of self-questions, questions similar to the original human questions can be generated. The proposed method works better than the two existing methods pipeline and end-to-end. The F1 criterion obtained during the code re-implementation process is evidence that the proposed model performed better than other methods.

The RoBERTa method used in this paper performs better than the BERT and BERTHAE methods in conversational Q&A. This method has been investigated on both QuAC and CANARD datasets and the results show that the value of F1 criterion on the QuAC dataset is higher than its value on the CANARD dataset.

The important point is that if the data is evaluated by human resources instead of using the model, a high F1 criterion will not be achieved; Hence, the HEQ evaluation criterion is used, which measures the performance of the model in relation to human responses to a question.

## 3 Methodology

In this process, in order to evaluate the accuracy of the claims mentioned in the paper and also the reusability of the paper codes, the codes introduced in GitHub were used. For this purpose, it was carried out in accordance with the instructions mentioned in it. Due to severe hardware limitations, it was not possible to teach the model and re-implement the code using Google Colab.

### 3.1 Model descriptions

For the Question Rewriting section, a trained T5 model is used to generate the word sequence. This model actually receives a question with a set of sentences as history and in return produces a self-contained question. CANARD data set has been used to teach and evaluate this model.

For the QA section, the RoBERTa model is used. The paper shows that the accuracy is much higher than other proposed models for this task, such as BERT and its fine-tuned version for answering the questions, ie. BERT+HAE.

BERT is a text word representation model that has been pre-trained on large collections. Although BERT is not designed for CQA, it works well on CQA databases. Receives text, current question, and previous conversation history as input.

BERT + HAE is a BERT-based QA model with a specific CQA module. embedding Answer history (HAE) is added to the embedding of BERT words. Using HAE, the answer information of previous questions is encrypted.

RoBERTa is an enhanced BERT using pre-training techniques to achieve strong optimized weights in larger bodies. In their experiments, the authors found that RoBERTa performs well in CQA and achieves performance comparable to the previous SOTA model, HAM (Qu et al, 2019b), in QuAC. Therefore, they used the RoBERTa model as their base model because of its simplicity and effectiveness.

### 3.2 Datasets

1- QuAC (Choi et al., 2018) contains 100,000 pairs of QAs in information retrieval conversations, where the student asks questions based on a topic with background information provided, and the teacher provides answers in the form of text sections in Wikipedia documents. For validation, they used a subset of the original QuAC training suite that included questions that matched their questions in the CANARD suite. The remaining data is used for training.

2- CANARD (Elgohary et al., 2019) consists of 31K, 3K and 5K QA binaries for training, validation and testing sets. Questions in CANARD are created by rewriting a subset of the main questions in QuAC. They used training and development kits to train and validate QR models and test kits to evaluate QA models.

The QaAC dataset can be downloaded from the link provided in GitHub. This dataset contains three separate files: train.json, valid.json and dev.json. The train.json file contains man-made questions, which is essentially the same as the QR rewrite model. If the model is retrained, the valid.json file is used to determine the optimal combination of hyperparameters and model evaluation.

### 3.3 Hyperparameters

This section explains the meta-parameters that have been used. To achieve QR and QA models, the transformers library and T5 and RoBERTa models have been used. The hardware used was the resources that Kolb made available to the public with 12GB of memory. For QA model, ADAMW optimizer with 3e-5 learning rate is used. The maximum length of the input sequence is 512 and the maximum length of the response is 128. 12 size children are used to teach RoBERTa. Of course, to reduce the memory consumption, we reduced this amount to one, but the problem was still not solved and the problem of lack of memory prevented RoBERTa retraining. Since we had a total of three loss functions, we needed to determine the effect of each on the final loss function with coefficients. For this purpose, using the values mentioned in the article, we considered the loss function coefficient of the QA section to be 0.5 and the coefficient related to the consistency loss function to be 0.6.

### 3.4 Experimental setup and code

After installing the requirements, the relevant packages and downloading the data set, the RoBERTa model taught in the article was used; By downloading and unziping from the relevant address. Using the F1 evaluation criterion, as well as setting the number of yarns to 20 and the number of batches to 100, the evaluation process was performed using the dev.json file.

### 3.5 Computational requirements

Contrary to what was mentioned in the article about executable hardware, Google Colab was used with the following specifications:

GPU 13GB RAM and 80GB Disk were used.

The duration of the model evaluation process on the Google Colab with the mentioned features is approximately 18 minutes and 30 seconds, and the disk and GPU consumed in this process are 49GB and 4GB, respectively.

## 4 Results

The F1 criterion obtained during the code reinforcement process is evidence that the proposed model performed better than other methods.

### 4.1 Results reproducing original paper

In the evaluation performed using the proposed RoBERTa model of the paper, the criterion F1 67.73891723166825 was obtained, which is quite similar to the accuracy reported by the paper itself. The main article also uses the HEQ-Q and HEQ-D criteria, which aim to evaluate the performance of the model in relation to humans, but here, due to the time-consuming nature of these evaluations, we were not able to achieve them. Because for a certain and relatively large number of questions, it is necessary to answer the questions manually and be measured by a higher level of monitoring than the model answers.

### 4.2 Results beyond original paper

In the mentioned article, for accurate evaluation between the previous methods (end-to-end and pipeline), all three ready-made models BERT, BERT + HAE and RoBERTa have been used. In this regard, it has been shown that the RoBERTa model gives better results for all three methods. Also, the proposed method was better accurate for each of the mentioned models than the other methods. The F1 benchmark was 1.2 for the QuAC dataset and 2.3 for the CANARD dataset.

## 5 Discussion

According to the comprehensive documentation provided, the article code was re-executed and evaluated in the colab context, and similar results were obtained to the results presented in the article. Due to hardware and GPU limitations, it was not possible to retrain the model on the other datasets mentioned in the article. Additional attempts and experiments were performed in this regard, including resizing batch and epoch, but to no avail.

### 5.1 What was easy

The hardware requirements and the initial setup of the experiment were fully described in the article in Section B (Hyperparameters), which was very helpful in re-executing the code.

A description of all usable datasets was also provided in Section 4.1 (Datasets). For implementation, the article was developed according to the documentation in the gate and only the QuAC dataset was used.

The documentation published at the gate by the authors was almost comprehensive and practical. These documents include installation requirements, hardware, data sets and their download addresses, how the model is taught (with all parameters), how the model is evaluated (with all parameters), as well as the evaluation criteria reported in the article (F1).

## 5.2 What was difficult

There were many hardware limitations to retrain data. The authors used a 24GB GPU (RTX TITAN). Execution with such conditions is not possible due to the free features provided by Google Colab.

Due to the mentioned limitation, we tried to change the batch size, which was set to 12 by default in the article, to 2; But we still had a lack of RAM from Kolb. It should be noted that by reducing the batch value, we also changed the number of epochs, but there were still problems, because at the beginning of the work and before the start of processing, it requires at least 24GB of RAM.

## 5.3 Communication with original authors

Due to the comprehensive documentation provided in the gate as well as in the text of the article, there was no need to interact with the authors. Of course, the gateway account and the authors' research gate account were available in ways to communicate with the authors, including email.

