# OpenReview forum: "Learn to Resolve Conversational Dependency: A Consistency Training Framework for Conversational Question Answering"
_ML_Reproducibility_Challenge/2021/Fall — Reject_

### Official Review · Reviewer_pQEx · 2022-02-25
**Short but detailed evaluation**

**Rating:** 7
**Confidence:** 3

**Review:**

The authors did what was expected. They reproduced the results given their hardware and matched the results. They did try it with different parameters and got similar results, which shows robustness. There was one case they weren't able to reproduce

---

### Official Review · Reviewer_NhMP · 2022-03-10
**Reproduction of the original authors' GitHub repo that is light on details about the scope**

**Rating:** 6
**Confidence:** 3

**Review:**

This is a review for the " Reproducibility report of Learn to Resolve Conversational Dependency: A Consistency Training Framework for Conversational Question Answering paper".

I went through the report, provided Jupyter notebook, original paper and the code repository from the original authors.  I first evaluate it using the ML Reproducibility Challenge - Reviewer Guidelines (https://docs.google.com/document/d/1O9QROqGmR4D2ncEj_suVpDzkcd3zdUAaDDjwyep5QTc/edit?pli=1) and then point out to some additional issues.

-   Reproducibility Summary : The authors provide a 1 page Reproducibility Summary.

-   Scope of reproducibility: The authors do not enumerate the scope of the reproducibility summary. For example, the report does not show the reproduced results for the BERT and BERT-HAE models, and for the CANARD dataset. This has not been explicitly stated neither in the report, nor the provided Jupyter notebook.

-   Code: The authors state that they reused the Github repository by the original paper authors (https://github.com/dmis-lab/excord). The provided notebook simply runs all instructions from the Github repo without further documentation or comments.

-   Communication with original authors: The report authors declare no communication with original authors but mention that communication channels were easily findable.

-   Hyperparameter Search: The report authors mostly used the hyperparameters provided by original paper authors, tweaking them slightly due to resource constraints. There is no evidence of further hyperparameter sweeping in the report and notebook.

-   Ablation Study: None.

-   Check if the report provides comprehensive ablation studies. Bonus if the ablations are backed up with proper motivation.

-   Discussion on results

-   Whether the report contains detailed discussion on the state of reproducibility of the original paper: The report authors mention that the original authors have provided comprehensive documentation for reproducing some results.

-   Recommendations for reproducibility: None.

-   Whether the authors provide any recommendations to the original authors for improving reproducibility: None

-   Results beyond the paper : Section 4.2: The report authors simply reiterate the results from the original paper. These claims have not even been reproduced in the provided notebook and are not beyond the claims of the original paper. The mentioned F1 benchmark numbers seem wrong and do not appear anywhere in the notebook. Line 122: "The F1 benchmark was 1.2 for the QuAC dataset and 2.3 for the CANARD dataset." These numbers refer to the improvements of the F1 score with the Excord framework (Section 4.5 in the original paper) and not absolute F1 scores. Moreover, these have not been reproduced in the notebook.

-   Overall organization and clarity: The paper would benefit from further proof-reading and addition of more details about the scope of the reproduction.

-   Grammatical issues / organization / proper plots
	-    Lines 31-33: "A Conversational Question Answering (CQA) is the design of an intelligent conversation system that can not only engage in an interactive conversation at the human level, but also go beyond it and answer questions on a variety of topics, one of which is today Prominent goals in the field of artificial intelligence." : "prominent" should be lower case, sentence needs restructuring.
	-    Lines 43-45: " In the pipe line method, to generate ambiguity questions from the main questions, if the T5 model is used to prepare a set of self-questions, questions similar to the original human questions can be generated." : ambiguous questions
	-    Lines 45-46: "The proposed method works better than the two existing methods pipeline end-to-end." - The proposed method works better than the two existing methods: pipeline and end-to-end.
	-    Lines 51-52: "The important point is that if the data is evaluated by human resources instead of using the model, a high F1 criterion will not be achieved;" : Sentence ends with a semi colon ;
	-    Lines 68-69: "embedding Answer history (HAE) is added to the embedding of BERT words. Using HAE, the answer information of previous questions is encrypted." HAE stands for history answer embedding, encrypted is not a suitable synonym for the word "encoded" in the original paper.
	-    Line 82: "The QaAC dataset can be downloaded from the link provided in GitHub." - Typo, should be QuAC.
	-    Line 97: "After installing the requirements, the relevant packages and downloading the data set, the RoBERTa model taught in the article was used; By downloading and unziping from the relevant address." : I am not sure if taught is the correct word for this context. Also, sentence should be reframed to make it gramatically correct.

Additional issues:
- Section 3.1 Model Descriptions have been copied straight from Section 4.2 of the original paper with some words changed to less applicable synonyms (encoded to encrypted, corpara to bodies).

- Section 3.2 does not mention the CoQA dataset, which is used in the original paper to demonstrate transfer learning (Section 4.1).

The train.json files contains questions generated by human annotators and questions generated synthetically by the QR model. The report states "The train.json file contains man-made questions, which is essentially the same as the QR rewrite model." This is a confusing claim and further clarification is needed about the nature of QR model generated questions.

- Section 3.3: Line 88: It is not clear what "Kolb" refers to. Seems to be a typo for Colab.

- Section 3.4: Line 99: It seems "yarns" is used as a synonym for threads. This is not correct usage for a technical term.

- Section 3.5: Mentioning the specific GPU used would be useful for future readers.

- Section 5.2: Line 141: Again, it is not clear what "Kolb" referes to.

---

The report authors provide a partial reproduction of the original paper. While verification of reproducibility is always beneficial, the authors should clarify which sections of the papers were skipped because of resource constraints. There is no mention of the reproduced performance of BERT and BERT+HAE models, nor on the CANARD dataset. The clarity of the report can also be greatly improved with additional proof reading.

---

### Meta-Review · Program_Chairs · 2022-04-07

**Recommendation:** Reject
**Confidence:** 4

**Metareview:**

A well-written submission with some interesting reproducibility results. However,  I agree with reviewer NhMP regarding the fact that the authors should clarify which sections of the papers were skipped because of resource constraints and there are some grammatical and spelling improvements to be done, and that some additional work should be dedicated for the report to be clearer regarding its contributions.

---

### Decision · Program_Chairs · 2022-04-09

Reject